# Abscisic Acid and Flowering Regulation: Many Targets, Different Places

**DOI:** 10.3390/ijms21249700

**Published:** 2020-12-18

**Authors:** Damiano Martignago, Beata Siemiatkowska, Alessandra Lombardi, Lucio Conti

**Affiliations:** Department of Biosciences, University of Milan, Via Giovanni Celoria, 26-20133 Milan, Italy; damiano.martignago@unimi.it (D.M.); beata.siemiatkowska@unimi.it (B.S.); alessandra.lombardi@unimi.it (A.L.)

**Keywords:** abscisic acid (ABA), flowering time, Arabidopsis, drought escape, drought, *bZIP*, *GIGANTEA*, *CONSTANS*, *FLOWERING LOCUS T*, *FD*

## Abstract

Plants can react to drought stress by anticipating flowering, an adaptive strategy for plant survival in dry climates known as drought escape (DE). In Arabidopsis, the study of DE brought to surface the involvement of abscisic acid (ABA) in controlling the floral transition. A central question concerns how and in what spatial context can ABA signals affect the floral network. In the leaf, ABA signaling affects flowering genes responsible for the production of the main florigen FLOWERING LOCUS T (FT). At the shoot apex, FD and FD-like transcription factors interact with FT and FT-like proteins to regulate ABA responses. This knowledge will help separate general and specific roles of ABA signaling with potential benefits to both biology and agriculture.

## 1. Introduction

Plant hormone signaling pathways are highly interconnected to allow plants to finely adjust growth and development according to varying environmental stimuli derived from growth conditions, nutrient availability, biotic and abiotic stress [1]. Abscisic acid (ABA) is long known to play central roles in drought, osmotic, and high salinity responses, hence generally considered as a stress-related hormone [2]. One of the best-characterized mechanism of action of ABA in response to drought stress is the control of transpiration via stomatal opening and closure [3]. However, there is a growing body of evidence that points to ABA involvement in plant growth and developmental processes well beyond stress responses. In well-watered, nonstressed conditions, ABA signaling is required in root tissues for growth, hydrotropism, xylem formation, and suberin deposition, in the leaves for leaf initiation and development (as reviewed in [4]). In this review, we will explore known and potential modes of interaction between ABA signaling and the genes that control the transition to flowering. To support the reader, we will introduce some specific notions related to the regulation of the floral transition, ABA biosynthesis and signaling while referring to more specialized readings whenever it will be required.

### Day Length Is a Key Floral Trigger in Arabidopsis

The transition to flowering marks the switch from the vegetative to the reproductive stage. Most plant species need to commit to flowering in a short window of time during the year to ensure optimal reproductive success. Hence, the timing of this transition is highly sensitive to environmental factors, enabling plants to align pollination, fruit and seed development with the most favorable conditions. Seasonal variations in mean temperature, day length (also known as photoperiod), and availability of nutrients and water are among the known environmental cues that regulate flowering.

Four floral pathways have been primarily described in the model plant *Arabidopsis thaliana* (Arabidopsis) through different genetic screens coupled with physiological analyses [5]. These pathways convey signals from photoperiod, vernalization, different endogenous cues as well as gibberellic acid (GA) accumulation, which is also linked to the age pathway [6]. Because of the established connections with ABA signaling, we will first focus on the photoperiodic pathway.

Most plants are sensitive to variations in photoperiod, which act as a critical seasonal cue at temperate latitudes [7]. In Arabidopsis, long day conditions promote flowering via activation of a signaling cascade that converges to the transcription of *FLOWERING LOCUS T* (*FT*) and *TWIN SISTER OF FT* (*TSF*) in the phloem companion cells [8]. Upon export from the phloem companion cells to sieve elements [9], FT and TSF gene products act as a florigenic signal, moving via the phloematic stream towards the shoot apex [10,11,12]. The photoperiodic cascade is activated upon exposure of plants to long-day conditions when GIGANTEA (GI) and the blue light receptor FLAVIN-BINDING, KELCH REPEAT, F-BOX1 (FKF1) display similar diel-regulated accumulations in the light phase [13,14]. The light-stabilized GI-FKF1 complex triggers the proteasomal degradation of CYCLING DOF FACTORs (CDFs), a family of transcriptional regulators that repress *CONSTANS* (*CO*) [15]. The diel degradation of CDFs alongside several mechanisms that regulate *CO* transcript accumulation contribute to a robust daytime expression of *CO* message, peaking at dusk [16]. Light is necessary to stabilize the CO protein, and numerous light-dependent molecular mechanisms involved in this process have been described [13]. CO protein plays a key role in the transcriptional activation of the florigens, forming a trimeric complex with NUCLEAR FACTOR-Y (NF-Y) B and C subunits at *CONSTANS RESPONSIVE* elements (CORE) located at the promoter of *FT* [17,18]. NF-Ys are highly conserved trimeric transcription factors (TFs) formed by NF-YA, NF-YB, and NF-YC subunits involved in many developmental processes including flowering [19]. Furthermore, *FT* transcript levels are further regulated by many transcriptional events that respond to a vast array of environmental and endogenous signals [20,21,22]. Thus, while typically CO is required for *FT* transcriptional activation, different TFs enable the fine-tuning of *FT* levels appropriate to the environmental conditions, thereby conferring substantial plasticity to the flowering process.

The mobilization of FT at the shoot apex triggers a change in the shoot identity that switches from producing leaves to floral primordia. As described in the current model for FT signaling, upon its relocation at the apex, FT protein forms a complex with the basic leucine zipper (bZIP) domain TF FD, probably with the participation of 14-3-3 proteins [23,24]. Although FD can bind DNA on its own, heterodimerization with FT (or TSF) and its phosphorylation enhance its DNA binding activity [24]. The function of FT is antagonized by a structurally related protein, TERMINAL FLOWER 1 (TFL1), which is also mobile, albeit its range of movement appears to be limited to the shoot meristem cells [25,26]. TFL1 can interact with FD and is recruited via FD at thousands of genomic positions where it exerts transcriptional repression [27,28]. FT outcompetes TFL1 for FD binding and thus activates transcription of FD targets which include floral meristem identity genes, conferring a floral fate to newly arising lateral primordia, and hormone-related gene functions [27,29].

In addition to the photoperiodic pathway, winter-annual accessions of Arabidopsis require the experience of cold to flower in the following spring, a process referred to as vernalization. This annual habit is conferred by two loci, the floral repressor *FLOWERING LOCUS C (FLC)*, a MADS-box type TF and its upstream activator *FRIGIDA* (*FRI*) encoding a coiled-coil domain protein acting as transcriptional regulator and chromatin modifier [30]. Exposure of vernalization-sensitive *FRI FLC* seedlings to cold temperature triggers the epigenetic silencing of *FLC*, which is mediated by several chromatin remodeling proteins [31]. The repressed state of *FLC* chromatin causes its transcriptional inactivation and is maintained through mitotic cell divisions upon a return to warm temperature. Misexpression studies allowed to define the spatial interactions between the photoperiod pathway and vernalization response. *FLC* represses *FT* in the leaf and several floral genes expressed at the shoot meristem including *FD* [32]. Thus, vernalization enables the transcriptional activation and mobilization of the main systemic flowering signal FT and its response in shoot meristem cells.

Flowering in Arabidopsis is also positively regulated by gibberellins which play an essential role under noninductive short-day conditions [33,34,35]. GA signaling is mediated by a class of proteins named DELLA that act as negative regulators of GA responses. DELLAs interact with a vast array of proteins (mainly TFs) which preside different hormonal and developmental processes [36]. According to a consolidated model, DELLA binding usually impairs TFs function or their DNA accessibility which blocks GA-regulated transcriptional events [37]. An increase in GAs cellular concentration triggers a signaling cascade that leads to DELLA ubiquitination and its proteasomal degradation, thus promoting TFs function. For these reasons, the GA pathway, via control of DELLA levels, plays a key integrative role by modulating multiple floral inputs in different spatial contexts [38].

## 2. ABA Signaling and Its Multiple Connections with the Photoperiodic Pathway

Water deficit conditions experienced by Arabidopsis during the vegetative phase result in accelerated flowering compared to normal watering conditions [39]. This plastic shift in flowering activation is considered adaptive and referred to as drought escape (DE), a bet-hedging strategy that enables plants to attain reproductive development and achieve an early seed set under water-scarce environments [40]. While succeeding in reproduction under potentially lethal drought conditions, the cost associated with this strategy is a considerable reduction in seed number production as a result of shortened vegetative growth [41].

Genetic screens identified several mutants impaired in DE, the vast majority of which are defective in the photoperiodic response [42,43]. Consistent with the requirement of long-day-stimulated photoperiodic signaling in DE activation, water deficit conditions applied under short days do not cause DE (conversely, they delay flowering). Accordingly, increased levels of florigen *FT* and *TSF* accumulate in response to water deficit only under long-day photoperiods. Thus, drought signals can be interpreted as positive cues for flowering depending on the activation status of the photoperiodic cascade. GI is required in this process, as no florigen expression occurs in *gi* mutants under any photoperiodic regime [39]. This initial model has been further refined to indicate that GI is not just indirectly required to activate the photoperiodic cascade (e.g., through the transcriptional activation of *CO*) [43]. Indeed, GI conveys drought-derived cues upstream of *FT* in parallel to CO (this aspect will be discussed in more detail below).

As the basic structure of the DE process is highly intertwined with the photoperiodic genes, the integration of drought stimuli with the floral network is in large part mediated by ABA. In flowering plants, ABA is synthesized via the carotenoid pathway by cleavage of β-carotene metabolites called xanthophylls and shares the same intermediate molecular pool of other plant hormones like cytokinins, brassinosteroids, and GA. The first steps of ABA biosynthesis take place in the plastid, with the oxidative cleavage of zeaxanthin into all-*trans*-violaxanthin by the enzyme zeaxanthin epoxidase, encoded in Arabidopsis by ABA DEFICIENT 1 (ABA1). The *NINE-CIS-EPOXYCAROTENOID DIOXYGENASES* (NCEDs) produce the C_15_ xanthoxin which is translocated from the plastid to the cytosol [44]. Xanthoxin biosynthesis is a rate-limiting step in ABA biosynthesis, hence NCEDs are major players in the regulation of ABA levels with specific developmental roles. *NCED3* is strongly upregulated by drought stress [45], and in concert with *NCED5* contributes to ABA-mediated drought stress responses [46]. From xanthoxin, bioactive ABA is synthesized in two steps. Firstly, ABA2 converts xanthoxin to abscisic aldehyde [47,48]. Secondly, Arabidopsis aldehyde oxidase 3 (*AAO3*) finally produces ABA from its aldehyde [49,50] in cooperation with a molybdenum cofactor encoded by the Arabidopsis *ABA3* gene [51,52].

Mutants of *aba1* and *aba2* are late-flowering under normal watering conditions [39,43]. Florigen transcript levels are also reduced in these ABA deficient mutants, which is associated with impaired DE compared to the wild type. Notably, the flowering time defect of ABA deficient mutants is restricted to long-day conditions, implying an interaction between ABA production and the photoperiodic response. *ABA2* expression occurs in the phloem companion cells, suggesting that these cells are a major source of ABA production [53]. Phloem-derived ABA may be translocated to other cell types including the shoot via specialized transporters or through the phloematic stream. Thus, while still unknown, the levels and distribution of ABA in the shoot might also affect florigen signaling beyond its site of production.

### Insights into the ABA-Flowering Crosstalk from the Analysis of ABA Signaling Mutants

The core ABA signaling cascade is composed of four main proteins and has been excellently reviewed elsewhere [54]. Briefly, ABA is bound by a family of soluble receptors known as PYR/PYL/RCARs [55,56]. Upon binding to ABA, PYR/PYL/RCARs interact with protein phosphatases (PP2C). This interaction inhibits the phosphatase activity of PP2Cs, allowing their substrate, protein kinases of the SNF1-RELATED PROTEIN KINASE 2 (SnRK2) group, to be phosphorylated. Active SnRK2s can, in turn, phosphorylate downstream components including bZIPs encoding ABA-responsive TFs/ABA-responsive element binding factors (ABFs/AREB) [57,58,59]. Phosphorylated ABFs enact the transcription of ABA/stress-response genes by direct binding on ABA-responsive elements (ABRE) on their promoter sequence [60,61]. In absence of ABA, PP2Cs bind SnRK2s, and keep them in a dephosphorylated, inactive form.

Several ABA signaling genes show expression in the vasculature [43,62]. Other than sharing similar spatial regulation with *FT*, ABA signaling mutants also display flowering defects that are consistent with a role in *FT* activation. Dominant alleles of the *PP2C ABI1* (*abi1-1*) encode proteins that are unable to dissociate from the SnRK2s even in the presence of ABA and thus impair ABA signaling [55,56]. Mutant *abi1-1* plants also fail to activate DE, which is associated with reduced levels of *FT*/*TSF* transcripts [43]. ABA exogenous applications activate flowering through the ABFs bZIPs which are classified in the same group A of FD-like bZIPs [63]. Interestingly, ABF3 phosphorylation on a LXRXX(S/T) motif, conserved among all ABFs, creates a 14-3-3 binding site [58]. In FD, disruptions in this C-terminal motif prevents FD function [28], and it has been observed in rice that the correct formation of this 14-3-3 binding site is required for the interaction of FD and FT homologs [23].

The *abf2*/*3*/*4* triple mutants show large alterations in the ABA-related transcriptome, including deregulation of *PP2C* genes, hinting to the possibility of a transcriptional feedback loop [64]. Additionally, triple *abf2*/*3*/*4* and quadruple *abf1*/*2*/*3*/*4* mutants display late flowering phenotypes, with reduced expression of *CO* and its transcriptional activator *FLOWERING BHLH 3* (*FBH3*) [57,65]. Notably, *abf3*/*4* mutants are late flowering under long-day conditions but not in short-day, and are impaired in DE compared with the wild type. The floral integrator *SUPPRESSOR OF OVEREXPRESSION OF CONSTANS 1* (*SOC1*) is a key target of ABF3 and ABF4 in the leaf. In turn, SOC1 indirectly promotes *FT*—but not *TSF*—expression by negatively regulating a set of *FT* repressors including *TEMPRANILLO1* (*TEM1*), *TEM2*, and *TARGET OF EARLY ACTIVATION TAGGED 1* (*TOE1*) encoding APETALA2 (AP2)-class transcriptional regulators. Interestingly, ABFs bind the *SOC1* promoter through the NF-Y complex by forming a direct interaction with NF-YC subunits. In triple *nf-yc3*/*4*/*9* mutants, the DE response is reduced, and *SOC1* transcription is unresponsive to ABA [63]. Beyond their role in the positive regulation of *FT* [17,66], NF-Y TFs are known to be important regulators of ABA-driven transcriptional responses [66]. The wide combinatory range offered by dimerization and trimerization of different NF-Y subunits, each one of these bearing unique functional domains [67], as well as their interaction with other ABA-regulated TFs like the ABFs [63], affects the specificity for DNA targets [68] and provides yet another layer of regulation in the crosstalk between ABA and flowering.

In apparent contrast, it has been reported that water deficit conditions can also repress flowering [39]. This response is observed under short-day conditions when the photoperiodic pathway is inactive. It is hypothesized that the drought-dependent repression of flowering occurs at the shoot meristem, acting independently or downstream of the florigen system. *FLC* plays a major contribution in this process, as mutants of *FLC* do not display delayed flowering in response to water deficit under short-day conditions. Another floral repressor, *SHORT VEGETATIVE PHASE* (*SVP*), a MADS-box type transcriptional regulator structurally related to *FLC* plays a central role in delaying flowering in response to water deficit under short-day conditions. Because FLC and SVP proteins physically interact, it is possible that these similar phenotypes reflect their mode of interaction and targets regulation. *SVP* transcript levels are upregulated in response to water deficit conditions, but not ABA applications [69]. On the other hand, *FLC* transcript levels increase in response to both water deficit and ABA applications [39,70]. Consistent with this ABA-*FLC* regulation, ABA-hypersensitive mutants (derived from loss-of-function alleles of multiple *PP2Cs*) display increased accumulation of *FLC* and are late-flowering compared to the wild type under short-day conditions. *abi1-1* mutants (which are ABA-insensitive) are early-flowering under short days and display reduced levels of *FLC* [43]. The TFs *ABA INSENSITIVE 4* (*ABI4*) encoding an AP2-class protein and the bZIP *ABA INSENSITIVE 5* (*ABI5*) were found to independently target *FLC* to promote its transcriptional activation in response to ABA [70,71].

An important question concerns how ABA levels or signaling may affect the GA-DELLA cascade. Recent data point to a general role for SVP in the control of ABA accumulation in leaves via negative regulation of ABA catabolism pathway genes *CYP707A1*, *CYP707A3*, and *AtBG1*. *svp* mutants display lower cellular ABA contents compared to the wild type and reduced drought stress tolerance [69]. There is also a known contribution of *SVP* at the shoot apex in the control of gibberellic acid biosynthesis, which plays a major role in the activation of flowering under noninductive conditions. SVP acts as a strong repressor of *GIBBERELLIN 20 OXIDASE 2*, encoding an enzyme required for GA biosynthesis. FT triggers the transcriptional repression of *SVP* at the shoot apical meristem (SAM), thus promoting GA accumulation [72]. Thus, variations in *SVP* levels caused by FT or water deficit can affect the ABA–GA balance globally or locally (i.e., in the shoot) to regulate flowering. ABI4 is another node of regulation of the ABA-GA homeostasis by activating the ABA biosynthetic gene *NCED6* and the GA catabolic gene *GIBBERELLIN 2 OXIDASE 7* [73]. ABA and GAs have opposite effects on ABI4 protein accumulation, positive and negative, respectively, indicating that water deficit conditions can alter the ABA–GA balance through modulation of ABI4 cellular abundance. Other than hormone production, the ABA–GA cross talk might occur at the signaling level. Recent studies in tomato indicate that DELLA acts in guard cells to promote stomatal closure, but this effect is ABA-dependent. Moreover, while DELLA in guard cells does not affect ABA levels, it increases guard cell ABA responsiveness [74]. While it is unknown whether this model can apply to Arabidopsis, it points to alternative modes of ABA–GA cross regulations possibly occurring at different tissue scales.

## 3. ABA Signaling Integration through GIGANTEA

*GIGANTEA* (*GI*) was identified as a key flowering gene, required for photoperiod perception and clock function. *GI* is also emerging as the key driver of DE, independent of its known role in the photoperiodic cascade. Given the multiple regulatory mechanisms coordinated by GI in the flowering regulatory process, it would be relevant to understand which step(s) could be sensitive to ABA levels. Here, we shall focus on the emerging role of GI in mediating hormonal signals (emphasizing the link to ABA signaling) and refer the reader to recent reviews detailing the mechanism of GI in photoperiodic and clock regulation [75].

Genetic evidence indicates that GI function is sensitive to ABA signaling status [43]. Impairing ABA signaling (as in *abi1-1*—mutants) causes marked reductions in *FT* and *TSF* accumulation even in a genetic background where *GI* is expressed constitutively via the *35S* promoter. Interestingly, *CO* levels are only moderately reduced in *35S::GI abi1-1* plants compared to *35S::GI*. This result supports a model where some aspects of GI protein function important for *FT*—but not *CO*—transcriptional regulation are sensitive to ABA signaling.

The idea that GI relay ABA signals onto *FT* with minor contributions from CO also derives from the study of *cdf1*/*2*/*3*/*5*/*gi* quintuple mutants characterized by high levels of *CO* transcript and an early flowering phenotype. These mutants failed to upregulate *FT* under water deficit conditions (as compared to *cdf1*/*2*/*3*/*5* quadruple mutants) and to activate DE, supporting the pivotal role of florigens in this process [43]. The interpretation of these results is that GI protein function is required to confer ABA-dependent responsiveness at the *FT* promoter.

The precise mode of GI-dependent florigen regulation promoted by ABA is still unclear. Drought or ABA alone cannot activate *FT* expression in *co* mutants and indicates that an interplay between GI and CO is ultimately necessary for *FT* activation and DE to occur. Interestingly, the florigen *TSF* can be transcriptionally activated in *co* mutants under water deficit conditions in a GI-dependent manner, indicating that in some cases, the interplay between GI and ABA is sufficient in promoting florigen expression [43]. This observation echoes the results of misexpression studies showing that GI can directly activate *FT* in the vasculature and partially rescue the late-flowering phenotype of *co* mutants [76]. GI is enriched at the *FT* promoter region, at positions usually occupied by strong *FT* repressors including SVP and the aforementioned TEM1/2, which are negatively regulated via the *ABF*/*SOC1* axis. The CDFs are also repressors of *FT*, and GI is required to relieve their repression at the promoter of *FT* [77] (Figure 1). Because GI does not present an obvious DNA binding domain, it may be recruited at the *FT* promoter through independent protein–protein interaction events to facilitate chromatin accessibility of positive regulators.

Chromatin immunoprecipitation (ChIP), followed by sequencing experiments, revealed that GI is recruited to the chromatin of thousands of loci to regulate gene expression. Interestingly, GI ChIP-seq peaks occur on regulatory regions of ABA/water deficit-related genes which are also differentially expressed genes in *gi* mutants [78]. GI was shown to interact with the ABA-related bZIP DC3 PROMOTER-BINDING FACTOR 4/ENHANCED EM LEVEL (DPBF4/EEL) to activate drought responses through the activation of the ABA biosynthetic gene *NCED3*. *eel*, *gi-1* and the corresponding double mutant have significantly lower expression of *NCED3* and present impaired stomatal closure in response to dehydration, thereby displaying a low survival rate in water deficit conditions [79]. These results could lead to a model where ABA-activated TFs recruit GI at different genomic positions to regulate gene networks related to drought stress. GI is found in complex with various enzymatic functions including kinases [80], O-fucosyltransferases [81], HEAT SHOCK PROTEIN 70/90 co-chaperones that promote the maturation of client protein interactors [82]. The most recent updates about potential and confirmed GI interactions can be found in [83]. The ever-growing list of interactors may suggest that GI acts as a scaffold protein, providing different enzymatic activities, possibly in conjunction with its recruitment at different genomic positions. Because GI protein levels oscillate during the day in a circadian manner, its recruitment to chromatin may gate DNA accessibility to TFs, and thus coordinate plant sensitivity to external signals in a diel manner [78]. Knowledge of these transcriptional mechanisms may help understand how GI can influence such a vast array of cellular responses.

## 4. Role of FD and FD-Like bZIPs Protein Complexes in Modulating ABA Signaling

An important theme arising from recent studies concerns the putative role of FD and FD-like proteins in the modulation of ABA signaling in complex with FT-like proteins. The Arabidopsis genome encodes 78 bZIP TFs, classified into 13 groups. These bZIPs have a basic domain required for the DNA binding activity and a characteristic leucine zipper domain that allows for homo- and heterodimerization [84]. The key floral genes *FD* and *FD PARALOGUE* (*FDP*) belong to group A of Arabidopsis bZIPs, totaling 13 members. Consistent with its established role in flowering, *FD* controls the expression of floral regulators like *SOC1*, *LEAFY*, *FRUITFULL*, and *APETALA1* [24,85,86,87] by direct binding at their respective promoters [87]. FD and FDP also share several downstream targets that are ABA- and water stress-related, including other members of group A bZIPs such as *ABF3* and *DPBF1*/*ABI5*, the *PP2Cs ABI1* and *HAB1*, the ABA catabolic gene *CYP707A2*, and proteostasis-related genes *ABI FIVE BINDING PROTEIN 2* (*AFP2*) and *AFP4* (Figure 1). *fd* and *fdp* mutant seedlings display reduced ABA sensitivity in germination assays [87]. Hence, one emerging aspect related to FD and FDP function is their role in the control of ABA response and metabolism. While the organization of these regulatory networks at the shoot apex is currently unclear, further confirmation for this interplay between FD and ABA-related genes derives from ChIP and expression analyses of the shoot-specific *TFL1* gene. The TFL1-FD complex directly represses, among others, the ABA biosynthetic gene *ABA1*, the bZIPs *ABF4*/*AREB2* and *ABI5* together with the *ABI5* regulator *AFP2* [27]. Thus, similar to floral targets, a competition between FT and TFL1 might modulate ABA levels or sensitivity in the shoot meristem cells (Figure 1).

During seed development, TFL1 was shown to stabilize ABI5 protein in the developing endosperm, possibly in response to ABA [88]. ABA involvement in seed development is known [89]. However, the newly discovered TFL1–ABI5 interaction further indicates multiple regulatory FD-like bZIP complexes that might have different roles and functions according to the tissue and developmental specific context in which these complexes form. ABI5 was described as a floral repressor, with transgenic plants overexpressing ABI5 showing delayed flowering under long-day conditions, owing to increased levels of *FLC* [70]. With the notable exclusion of FD and FDP, most of the molecular events involving the Arabidopsis group A bZIPs were studied in seedlings, and little is known about their targets at the SAM during floral transition. However, many of the characteristics of these ABA related bZIPs—notably, their ability to heterodimerize, to bind to 14-3-3 proteins, and their structural similarity with FD and FDP—could hint to a more prominent role for this protein family at the apex. It is possible to hypothesize a highly fluid and dynamic model acting at the SAM. Different bZIPs can be activated by the relocation of FT and TSF to the SAM, and act in cooperation or antagonistically with FD and FDP, to integrate photoperiodic and hormonal signals (Figure 1). Different combinations of group A bZIP heterodimers could further allow or deny protein–protein interactions [90], and ultimately target different subsets of downstream components of flowering or ABA-related signal cascades (Figure 1).

## 5. Molecular Insights into the ABA–Flowering Relationship in Crops

While crop production is facing the threat of climate change, with extreme meteorological drought predicted to be more common [91], DE response could either be either beneficial or maladaptive, depending on the drought scenario [92]. Maintaining a prolonged vegetative growth could give a competitive advantage in terms of seed number, while DE and a short life cycle can increase reproductive success in drought-prone environments at the expense of productivity and yield under sufficient watering conditions [93,94]. Taking into account our continuous progresses in deciphering the ABA–flowering molecular interactions in Arabidopsis, a key question is how this knowledge can be translated to other species, including crops. Valuable information may derive from the study of DE traits in natural populations as shifts in flowering time phenology is a major trait enabling their survival [95,96]. This wealth of knowledge will lead into molecularly exposed allelic variations that confer plasticity in DE and can help uncoupling flowering responses from generic drought and abiotic stress responses, providing novel breeding and biotechnological targets for crop improvement.

In the model monocot crop rice (*Oryza sativa* L.), domesticated at tropical latitudes, flowering is induced by the transition from long to short day conditions [97], while in temperate rice varieties, flowering is photoperiod-insensitive [98]. Interestingly, the DE pathway is conserved in its essential components in rice. ABA is required to activate the DE pathway in rice, with ABA-deficient rice mutants being impaired in the DE response. *OsGIGANTEA* (*OsGI*) is involved in the DE response, and *OsGI-RNAi* lines present reduced DE response, albeit its role appears to be ABA-independent [99]. In apparent contrast, severe drought stress delays flowering in rice, with the repression of the rice florigens *HEADING DATE 3A* (*Hd3a*) and *RICE FLOWERING LOCUS T 1* (*RFT1*) [100]. Delaying flowering under drought stress could also be mediated by the rice floral repressor *RICE CENTRORADIALIS 1*, a TFL-like gene induced by ABA under severe stress conditions [101]. Following a biotechnological approach, Miao et al. [102] used gene editing techniques to obtain rice mutants that are less sensitive to ABA by mutating multiple genes of the ABA receptor *PYR*/*PYL*/*RCAR* family. These lines had higher growth rates and improved yields compared to those of the wild type (∼+25%), but also displayed increased sensitivity to water stress conditions. Higher productivity was in part determined by an extended duration of the vegetative phase. Unlike *OsPYL* loss-of-function mutants, overexpression of *OsPYL/RCAR5* improved salt and drought tolerance in rice during the vegetative growth stage. In contrast, in normal watering conditions, seed yield was greatly reduced (∼−75%) [103], pointing to a tradeoff between growth duration and drought tolerance traits. Studies with introgression lines led to the discovery that most drought tolerance-associated quantitative trait loci (QTLs) are independent of DE-QTLs, pointing at the evolution of at least two distinct adaptive strategies in rice under drought stress [104]. DE seems an effective strategy that might contribute to improving yield under stress. However, this contribution varies depending on specific drought scenarios, and on the developmental stage in which drought stress is imposed [105]. Collectively, these results imply a significant and yet uncharacterized contribution of ABA in the control of the floral transition of rice, whereby DE activation depends on the genotype and the intensity and timing of the drought stress.

Little information is available about ABA molecular control of flowering in other monocots, however, in a pioneeristic attempt to produce drought-resistant crops, transgenic maize plants constitutively expressing *ZmNF-YB2* showed a range of ABA-related developmental responses, including higher stomatal conductance and chlorophyll content and delayed onset of senescence, leading to improved yield under severe drought conditions. These advantages under drought conditions were highly situational [106]. Fast cycling, early flowering crops could avoid terminal drought and reduce the length of the crop season, but often this strategy pays a cost in terms of reduced yield [107,108]. It is still unclear what genetic adjustment may be needed to manipulate ABA sensitivity and flowering time. Resolving the flowering-specific effects of ABA from its general role in drought stress response may lead to improvements to crop yield whilst maintaining stress responsiveness in specific environments.

## Figures and Tables

**Figure 1 ijms-21-09700-f001:**
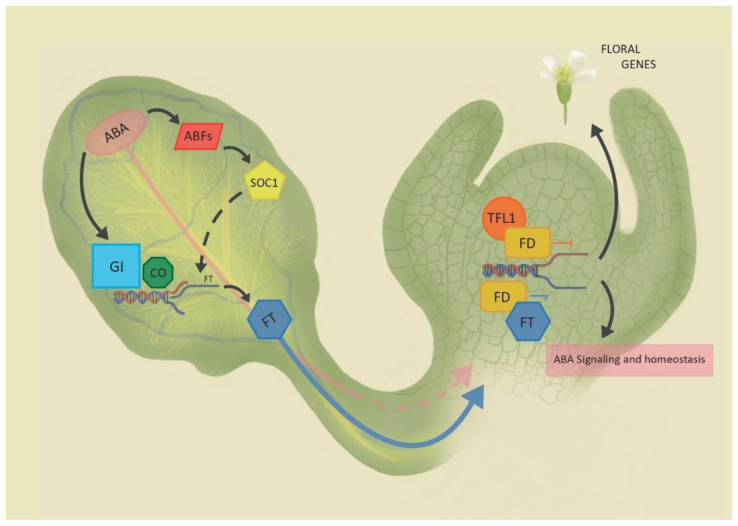
Abscisic acid (ABA) signaling and flowering regulation. In the leaves (left), ABA controls FLOWERING LOCUS T (FT) transcription acting on GIGANTEA (GI) and CONSTANS (CO); ABA-responsive transcription factors (ABFs) can modulate SUPPRESSOR OF OVEREXPRESSION OF CONSTANS 1 (SOC1) expression, in turn affecting FT transcription through an indirect mechanism. FT moves to the SAM where it interacts with FD and FD-like basic leucine zippers (bZIPs) to activate floral genes and ABA signaling transcriptome. ABA is transported in the phloem, but its roles at the SAM are not yet known. TERMINAL FLOWER 1 (TFL1) antagonizes FT, repressing transcription. Dashed lines represent indirect or not yet confirmed pathways, while full lines represent known ones.

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
