# Peer review of "Abscisic Acid and Flowering Regulation: Many Targets, Different Places"

_ijms, 2020, doi:10.3390/ijms21249700_

Round 1

Reviewer 1 Report

This review is very pleasant to read and synthetizes the role direct or indirect of the ABA pathway in the floral transition. 

I have some questions and propositions listed below 

  • I find that the only figure in the manuscript is not very informative and could be improved ; moreover other figures should be proposed to help the reader to well understand all pathways interactions (for example Khan et al 2014)
  • the authors present almost exclusively data from Arabidopsis: there are lot papers on others plants and notably food plants -some examples or a table would allow the manuscript to be expanded. 
  • a scheme of ABA pathway is necessary because the heart of the manuscript

Author Response

We would like to thanks the reviewer for the comments and suggestions.

We decided to keep the figure as simple as possible to deliver a concise message to the reader. Including all possible interactions between ABA and flowering time would make any figure hard to decipher.

We substantially expanded the final section to include a contribution to the field from rice studies (see lines 330-350). It is true that there is a consistent body of science coming from crop and non-model species, but we focussed on the molecular regulatory pathways and Arabidopsis is by large the better characterized model. For clarity, we changed the subtitle of the last section in our review.

With the short time available, we could not be able to provide more high-quality figures and schemes. Regarding ABA signalling, we referred to a number of reviews that provide detailed schemes for biosynthesis and signalling ABA pathways. Thanks for your understanding.

Reviewer 2 Report

            The review by Martignano et al summarizes recent advances in the regulation of flowering time by abscisic acid. This manuscript expands previous reviews from the group (Conti, 2017; Conti, 2019) with updated information about a process of interest for researchers working on plant development and/or plant stress responses. The manuscript is well written and structured. I only miss more discussion about potential connections of ABA and gibberellins signaling in the regulation of flowering time. I consider the review suitable for publication at IJMS. I just have a few minor recommendations and corrections as detailed below:

- Acronym for CYCLIN DOF FACTORs must be provided in line 56

- Line 61: it should be indicated that NF subunits are transcription factors

- Lines 68-79: The sentences “Among the FT-FD complex targets are different floral meristem identity genes that confer a floral fate to newly arising lateral primordia” and “FT outcompetes TFL1 for FD binding and thus activates transcription of FD targets which include floral genes and hormone-related gene functions” are partially redundant and could be fused in one sentence at the end of the corresponding paragraph

- Line 147: “vasculature tissue” should be substituted by “vascular tissue” or “vasculature”

- Line 153: I suggest to rewrite this paragraph, or to fuse it to the previous one, as “The ABFs are key components of ABA signaling, as they enact the transcription of ABA/stress-response genes by direct binding on ABA-responsive elements (ABRE) on their promoter sequence [52,53] upon phosphorylation by SnRK2s [54–56]” has been commented previously (lines 143-146)

- Line 243, the sentence “GI competes against PHYTOCHROME INTERACTING FACTOR 3 (PIF3) for the same binding regions on a genome-wide scale” looks irrelevant for the processes described in the review

- Line 249: “are impaired stomatal closure” should be corrected

- Line 265, another redundancy that should be corrected: “These bZIPs have a basic DNA-binding domain required for the DNA binding activity…”

- Line 285: “FT-like” needs to be substituted by “FD-like”

- Line 322: “shown” should be corrected by “showed”

- Line 327: “It still unclear” should be corrected by “It is still unclear”

- The format of some bibliographic references needs to be checked

- The abbreviations list should be checked. Many abbreviations included in the text are not listed

Author Response

We would like to thanks the reviewer for the detailed comments and suggestions. As suggested, we expanded the discussion about ABA and GI interactions providing some potential crosstalk mechanisms (lines 182-201).

We addressed the recommendations as commented below each point:

- Acronym for CYCLIN DOF FACTORs must be provided in line 56

  • done

- Line 61: it should be indicated that NF subunits are transcription factors

  • A descriptive sentence has been added

- Lines 68-79: The sentences “Among the FT-FD complex targets are different floral meristem identity genes that confer a floral fate to newly arising lateral primordia” and “FT outcompetes TFL1 for FD binding and thus activates transcription of FD targets which include floral genes and hormone-related gene functions” are partially redundant and could be fused in one sentence at the end of the corresponding paragraph

  • Thanks for the suggestion: the sentences were fused into “FT outcompetes TFL1 for FD binding and thus activates transcription of FD targets which include floral meristem identity genes, conferring a floral fate to newly arising lateral primordia, and hormone-related gene functions”

- Line 147: “vasculature tissue” should be substituted by “vascular tissue” or “vasculature”

  • Thanks

- Line 153: I suggest to rewrite this paragraph, or to fuse it to the previous one, as “The ABFs are key components of ABA signaling, as they enact the transcription of ABA/stress-response genes by direct binding on ABA-responsive elements (ABRE) on their promoter sequence [52,53] upon phosphorylation by SnRK2s [54–56]” has been commented previously (lines 143-146)

  • paragraphs 143-146 and 153-157 has been slightly modified to improve readability

- Line 243, the sentence “GI competes against PHYTOCHROME INTERACTING FACTOR 3 (PIF3) for the same binding regions on a genome-wide scale” looks irrelevant for the processes described in the review

  • sentence removed

- Line 249: “are impaired stomatal closure” should be corrected

  • “are” changed in “present”

- Line 265, another redundancy that should be corrected: “These bZIPs have a basic DNA-binding domain required for the DNA binding activity…”

  • redundancy corrected, thanks

- Line 285: “FT-like” needs to be substituted by “FD-like”

  • corrected, thanks

- Line 322: “shown” should be corrected by “showed”

  • corrected, thanks

- Line 327: “It still unclear” should be corrected by “It is still unclear”

  • corrected, thanks

- The format of some bibliographic references needs to be checked

  • bibliography has been checked carefully

- The abbreviations list should be checked. Many abbreviations included in the text are not listed

  • we extended the abbreviation list. In addition, all the abbreviations used in the text are defined at the first use with acronym in parenthesis